# Morphology, Morphogenesis and Molecular Phylogeny of Two Freshwater Ciliates (Alveolata, Ciliophora), with Description of *Pseudosincirra binaria* sp. nov. and Redefinition of *Pseudosincirra* and *Perisincirra*

**DOI:** 10.3390/microorganisms12102013

**Published:** 2024-10-04

**Authors:** Lijian Liao, Yue Hu, Xiaozhong Hu

**Affiliations:** 1Key Laboratory of Mariculture (Ministry of Education), College of Fisheries, Ocean University of China, Qingdao 266003, China; liaolijian@stu.ouc.edu.cn; 2Key Laboratory of Evolution and Marine Biodiversity (Ministry of Education), Institute of Evolution and Marine Biodiversity, Ocean University of China, Qingdao 266003, China; 3School of Biological Sciences, Reading University, Reading RG6 6AH, UK; hannah.hu@student.reading.ac.uk

**Keywords:** 18S rRNA gene, Deviatidae, infraciliature, new species, phylogenetic position

## Abstract

Ciliated microeukaryotes are insufficiently investigated despite their ubiquity and ecological significance. The morphology and morphogenesis of a new Stichotrichida species, *Pseudosincirra binaria* sp. nov., and the known *Perisincirra paucicirrata* Foissner et al., 2002, are here studied using live observations and protargol staining methods. The new species is characterized by having one buccal, one parabuccal and three frontal cirri, one frontoventral row extending to the posterior half of the cell, three left and two right marginal rows and three dorsal kineties with the left kinety conspicuously bipartite, along with one caudal cirrus at the rear end of each kinety. During morphogenesis, there exist five frontal-ventral cirral anlagen with anlagen IV and V forming a frontoventral row in the proter, and four cirral anlagen with only anlage IV generating a frontoventral row in the opisthe. The anlagen for marginal rows and dorsal kineties develop intrakinetally. The new population of *Perisincirra paucicirrata* corresponds well with other isolates regarding morphology and cell development. Phylogenetic analyses based on small subunit ribosomal gene sequence data revealed that both *Perisincirra* and *Pseudosincirra* are deeply clustered in the clade consisting of species from the genera *Deviata* and *Heterodeviata*, supporting the placement of both genera into the family Deviatidae Foissner, 2016.

## 1. Introduction

As an important component of aquatic ecosystems, ciliates (Alveolata, Ciliophora) are among the most species-diverse groups of unicellular eukaryotes occurring in various habitats, including freshwater, soil and seawater [1,2,3,4,5,6,7,8,9,10,11,12,13,14,15,16,17,18]. Class Spirotrichea Bütschli, 1889, the most complex and highly differentiated taxon within the phylum Ciliophora Doflein, 1901, has an extremely high species diversity with over 1200 nominal species currently identified [7,8,9,19,20,21,22,23,24,25,26,27,28,29,30,31,32,33,34,35,36], some of which can be assigned into the insufficiently studied order Stichotrichida Fauré-Fremiet, 1961 with eight families included so far. Stichotrichids can be recognized by having an ellipsoidal body shape, variable size from small to large, and ventral cirri arranged in one or more longitudinal and linear rows (not zig-zag as in Urostylida). Despite the recent documentation and characterization of several new taxa, a significant number of species await discovery due to under-sampling, leading to numerous confusions in the classification and phylogeny of stichotrichids [24,28,29,37].

Foissner (2016) established the new family Deviatidae to comprise *Deviata* Eigner, 1995, *Notodeviata* Foissner, 2016, and *Idiodeviata* Foissner, 2016 [7,34]. Thereafter, three more genera were added to the family [24,28]. These deviatid genera share two common features, viz. several longitudinal cirral rows dividing intrakinetally or with multiple within-anlagen and the absence of transverse cirri. Sequence information is currently available for only ten known species within the Deviatidae, and morphogenesis data are limited to several species from the genera *Deviata* and *Heterodeviata*, highlighting the need for further investigations to clarify the biodiversity of this taxon.

The genus *Perisincirra* was erected by Jankowski (1978) with *P. kahli* (syn. *Uroleptus kahli* Grolière, 1975) as the type species [35]. Berger (2011) provisionally assigned *Perisincirra* to the family Kahlillidae Tuffrau, 1979, and defined this genus as follows: adoral zone of membranelles and even undulating membranes (endoral and paroral) roughly following the pattern of *Gonostomum*; two or more cirral rows both to the right and left of the midline, with the cirri within these rows being very widely spaced; three frontal cirri, buccal and parabuccal cirrus/cirri present, but a frontoterminal, postoral ventral, pretransverse and transverse cirri absent; three or four dorsal kineties with possible caudal cirri at rear ends; body very flexible and contractile; two macronuclear nodules; lack of cortical granules; cirri rather fine and long but dorsal bristles short (about 3–5 μm) [20]. Gao et al., (2021) [24] transferred *Perisincirra longicirrata* Foissner et al., 2002 to *Pseudosincirra* Gao et al., 2021 [24] and classified it into Deviatidae. Consequently, *Perisincirra* currently comprises only two species, i.e., *Perisincirra kahli* (Grolière, 1975) Jankowski, 1978, and *Perisincirra paucicirrata* Foissner et al., 2002 [24]. Gao et al., (2021) defined *Pseudosincirra* as deviatids having a dorsal ciliature in *Urosomoida* pattern and caudal cirri and considered the inner two cirral rows to right of midline in *Pseudosincirra longicirrata* to be frontoventral rows [24]. However, Li et al., (2013) judged that the inner one or two right rows of *Perisincirra kahli* and *Pseudosincirra longicirrata* are probably marginal rows [36]. Based on the description and illustration of Gao et al., (2021) [24], we infer that perhaps these authors misinterpreted frontoventral rows. Therefore, we believe that the genus *Pseudosincirra* possesses one long frontoventral row and two right marginal rows, which is consistent with the research of Li et al., (2013) [36]. In view of the fact that neither *Perisincirra kahli* nor *P. paucicirrata* has a frontoventral row, whereas *Pseudosincirra longicirrata* does, we believe that the genera *Perisincirra* and *Pseudosincirra* should be differentiated on the basis of the presence or absence of a frontoventral row rather than the number of right marginal rows. Therefore, the diagnosis of *Pseudosincirra* and *Perisincirra* should be amended (see Section 3 below).

To explore the diversity of stichotrichid ciliates, especially little-known deviatids, we conducted a faunistic study at Qingdao during the last two years. In the present work, the morphology and morphogenesis of a new species, *Pseudosincirra binaria* sp. nov. and a new population of *Perisincirra paucicirrata* were described. Furthermore, their phylogenetic relationships were analyzed based on the newly obtained 18S rRNA gene sequences.

## 2. Materials and Methods

### 2.1. Sample Collection, Observations, and Identification

*Pseudosincirra binaria* sp. nov. was collected on 12 May 2023 from an artificial freshwater pond in Qingdao, China (36°03′03″ N, 120°21′06″ E). The water temperature was 18 °C at the time of sampling. *Perisincirra paucicirrata* was isolated on 6 May 2023 from a water sample collected from a freshwater pond in Baihuayuan Garden, Qingdao, China (36°03′55″ N, 120°20′44″ E). The water temperature was 20 °C at the time of sampling. The raw cultures were maintained at room temperature (approximately 25 °C) in Petri dishes, utilizing habitat water with squeezed rice grains to enhance the growth of bacterial food for the ciliates.

Observations of living cells were conducted using bright field and differential interference contrast illumination with light microscopes (Zeiss AXIO Imager D2, Jena, Germany and Olympus BX53, Tokyo, Japan). Protargol staining following the method of Wilbert (1975) was applied to reveal the infraciliature and nuclear apparatus [38]. In vivo measurements were performed at a magnification of 40–1000×. Measurements and counts on stained specimens were carried out at a magnification of 1000×. Drawings of stained specimens were carried out based on micrographs. To illustrate the changes occurring during morphogenesis, old (parental) structures are shown in outline, while new structures are shaded black. Classification follows Lynn [2] and Foissner [7], and terminology follows Berger [20].

### 2.2. DNA Extraction, PCR Amplification, and Gene Sequencing

Based on morphological study mentioned above, single cells of each species were isolated from a raw culture (without other deviated species recognized) and underwent five washes with sterilized bi-distilled water to eliminate potential contaminants. Subsequently, each cell was transferred to a 1.5 mL micro-centrifuge tube with a minimal volume of water. DNA extraction was performed using the DNeasy Blood and Tissue Kit (Qiagen, Germany), following the manufacturer’s instructions. PrimeSTAR Max Premix (2×) DNA polymerase (Takara) was utilized to minimize experimental errors occurring due to the polymerase chain reaction (PCR). The PCR primers 18S-F (5′-AACCTGGTTGATCCTGCCAGT-3′) and 18S-R (5′-TGATCCTTCTGCAGGTTCACCTAC-3′) were used for the amplification of 18S rRNA gene according to Medlin [39]. PCR conditions consist of an initial denaturation step at 98 °C for 30 s, followed by 35 cycles of denaturation at 98 °C for 10 s, annealing at 55 °C for 15 s, and extension at 72 °C for 10 s. The PCR products were sequenced bidirectionally by the Sangon Biological Technology Company (Qingdao, China). The contigs of each species were assembled using Seqman software ver. 7.1.0 (DNA Star), and the poorly sequenced bases at the beginning were removed based on peak profiles.

### 2.3. 18S rRNA Gene Sequence and Phylogenetic Analyses

The two newly obtained sequences were aligned with 79 other taxa sequences obtained from the National Center for Biotechnology Information (NCBI) GenBank database using MUSCLE on the EBI website (http://www.ebi.ac.uk/Tools/msa/muscle/, accessed on 28 June 2024). Accession numbers were provided to the right of species names in the phylogenetic tree. Four euplotids species, namely *Apodiophrys ovalis* (GU477634), *Diophrys scutum* (JF694040), *Paradiophrys zhangi* (FJ870076), and *Uronychia multicirrus* (EU267929), were used as the out-group taxa. Subsequently, primer sequences were manually removed from the alignment using Bioedit 7.2.5 according to Hall [40]. Both ends of the alignment were manually trimmed, resulting in a final alignment with 1758 sites, which was then used for constructing the phylogenetic trees.

Maximum likelihood (ML) analysis was conducted using IQ-TREE v.2.0 with 10,000 ultrafast bootstrap replicates. The TIM2 + F + R3 model was selected as the best-fit model based on the Bayesian information criterion (BIC) [41]. Bayesian inference (BI) analysis was performed using MrBayes ver. 3.2.6 on the CIPRES Science Gateway (XSEDE v.3.2.6) [42], with the GTR + I + G model chosen by Akaike Information Criterion in MrModeltest ver. 2.2 [43]. Four Markov chain Monte Carlo (MCMC) simulations were run for one million generations with a sampling frequency of 100 and a burn-in of 25,000 trees. The remaining trees were used to generate a consensus tree and calculate posterior probabilities according to the majority rule. The topologies of phylogenetic trees were visualized by MEGA X [44] and Seaview v.5.0.5 [45]. ML bootstrap values ≥95%, 71–94%, 50–70% and <50% were considered as high, moderate, low and no support, respectively [46]. Bayesian posterior probabilities ≥0.95 and <0.95 were considered as high and low support, respectively [47].

### 2.4. ZooBank Registration

The ZooBank registration number of present work is: urn:lsid:zoobank.org:pub: C1FC2E71-D2B1-4FB6-B3AD-962331C521DD.

## 3. Results

Class Spirotrichea Bütschli, 1889 [48]

Order Stichotrichida Fauré-Fremiet, 1961 [49]

Family Deviatidae Foissner, 2016 [7]

Genus *Pseudosincirra* Gao et al., 2021 [24]

Improved Diagnosis

Medium-sized non-dorsomarginalian Deviatidae with frontal, buccal and caudal cirri present. Two or more marginal rows on each side and one frontoventral row. Three dorsal kineties. The oral primordium originates apokinetally between right and left cirral field. Four or five frontal-ventral cirral anlagen.

### 3.1. Pseudosincirra binaria sp. nov.

ZooBank registration

*Pseudosincirra binaria* sp. nov.: urn:lsid:zoobank.org:act: 12F68041-ECDD-4EC1-A4AF-856173605E6F.

#### 3.1.1. Diagnosis

Freshwater species with a life size of 125–160 × 30–55 μm and an elongated elliptical to bluntly fusiform body shape; contractile vacuole positioned at about 45% of cell length; four macronuclear nodules and two micronuclei. Adoral zone extending approximately one-fourth of body length in vivo, composed of on average 23 membranelles; paroral and endoral membrane straight and optically parallel to each other. Consistently three frontal cirri, one buccal cirrus, one parabuccal cirrus, one row of about 18 frontoventral cirri, and three left and two right marginal rows; one caudal cirrus at the posterior end of each dorsal kinety, and the left dorsal kinety bipartite.

#### 3.1.2. Type Locality

An artificial freshwater pond (36°03′03″ N, 120°21′06″ E), Qingdao, China. The water temperature was 18 °C.

#### 3.1.3. Etymology

The species-specific name *binaria* is composed of Latin adjective *binarius* (two part) and the thematic vowel *a*-, alluding that the left dorsal kinety is bipartite.

#### 3.1.4. Type Materials

One slide (registration number: LLJ2023051204/1) containing the holotype specimen (circled with black ink on the back of the slide; Figure 1B–E) and forty-three paratype slides (registration numbers: LLJ2023051204/2–44) with protargol-stained morphostatic and dividing specimens have been deposited in the Laboratory of Systematic Taxonomy, Ocean University of China, China. Another slide with protargol-stained specimens (registration number: LLJ2023051204/44) has been deposited in the Marine Biological Museum, Chinese Academy of Sciences, Qingdao, China.

#### 3.1.5. Ecology

The new species is a very voracious consumer and grows well in Petri dishes at room temperature with freshwater medium to which a few rice grains were added to stimulate the growth of bacteria.

#### 3.1.6. Description

Cell size highly variable, 125–160 × 30–55 μm in vivo (*n* = 13), usually about 145 µm × 40 µm, ratio of length to width ranging 2.8–4.2:1 (on average 3.7:1) from freshly collected samples; when well-fed, some ciliates become bigger (Table 1). Body flexible but non-contractile, elongated elliptical to bluntly fusiform, with both ends slightly narrowed and rounded (Figure 1A,F,G). Constantly four macronuclear nodules situated left of midline. Mostly two oval micronuclei, about 3.0 μm in diameter after protargol staining, one located between the anterior two macronuclear nodules and the other located between the posterior two macronuclear nodules (Figure 1A,C,E,K,O). Contractile vacuole positioned dorsally close to left cell margin at about anterior 45% of the body length, about 11 μm across, pulsating at intervals of about 30 s (Figure 1A,F,K). Cytoplasm grayish, containing numerous globular crystals and many lipid droplets, which render cell opaque and dark at low magnifications (Figure 1F,G,I,J). Locomotion mainly by swimming or crawling on the substrate.

Buccal field narrow and inconspicuous, occupying about one-fourth of body length in vivo (in a range of 20–31%) and on average 24% of body length (in a range of 19–32%) after protargol staining (Figure 1A,B,D,H,L,M). Adoral zone of membranelles roughly in *Gonostomum*-pattern, composed of 20–25 membranelles (Figure 1B,D,L–N; Table 1). Paroral and endoral membranes almost straight, nearly the same length and both generally single-rowed, optically side by side but rarely staggered (1 out of 25 individuals examined) (Figure 1B,D,N). Pharyngeal fibers conspicuous, about 37 μm long (Figure 1B,D,L). Constantly three slightly enlarged frontal cirri; one buccal cirrus right of anterior of paroral membrane. One parabuccal cirrus behind the right frontal cirrus (Figure 1B,D,L–N). One long frontoventral row, with 15–21 cirri, commences slightly behind right of the parabuccal cirrus and terminates roughly at the same level as the penultimate cirrus of the inner left marginal row (Figure 1B,D,M). Consistently three left marginal rows, both the inner and outer row with 14–19 (on average 17) cirri, the middle row with 14–18 (on average 16) cirri; two right marginal rows with about 23 cirri each (Figure 1B,D). Marginal cirri about 12 μm long in protargol-stained specimens (Figure 1D,P). All cirri in frontoventral and marginal rows fine and widely spaced, each consisting of four basal bodies.

Invariably three dorsal kineties, dorsal bristles about 4.5 µm long in preparations (Figure 1E,P); the left kinety with 7–10 (on average 9) dikinetids, in the middle part with a conspicuous gap of 60 μm wide; the middle and right kineties almost bipolar, with 16–20 (on average 18) and 12–16 (on average 14) dikinetids, respectively (Figure 1C,E). Constantly three caudal cirri, one at the posterior end of each dorsal kinety (Figure 1C,E,P).

#### 3.1.7. Morphogenesis During Binary Fission

##### Stomatogenesis

In early dividers, stomatogenesis commences with the de novo formation of opisthe’s oral primordium (OP) in the postoral area between the frontoventral row and inner left marginal row (Figure 2B and Figure 3B). Subsequently, with the proliferation of basal bodies, the OP lengthens and differentiates into new adoral membranelles for the opisthe in its right anterior portion (Figure 2C–F and Figure 3C–F). Simultaneously, undulating membranes anlage (frontal-ventral cirral anlage I) is detached from the right side of the OP (Figure 2E and Figure 3E). Then, the posterior majority of anlage I splits longitudinally to form paroral and endoral membranes (Figure 2I and Figure 3I). With further development, the formation of all new membranelles is gradually completed from anterior to posterior in opisthe. The parental adoral zone of membranelles remains intact and is wholly inherited by the proter, while the parental undulating membranes begin to disintegrate and form anlage I of the proter (Figure 2N,O and Figure 3N,O), which then develops in the same way as in the opisthe.

##### Ventral Ciliature

Frontal-ventral cirral anlagen II and III evidently appear later than OP. Anlagen II and III occur earlier in opisthe compared to proter. In opisthe, anlagen II and III originate from the OP, of which II appears earlier than III (Figure 2E–G and Figure 3E–G); whereas in proter, they are derived from the dedifferentiation of buccal cirrus and parabuccal cirrus, respectively (Figure 2N and Figure 3N). When the new adoral zone of membranelles are almost completely formed, anlage IV emerges within the frontoventral row in both proter and opisthe (Figure 2P and Figure 3P); during this period, anlage V of the proter appears within the inner right marginal row (Figure 2Q and Figure 3Q) and subsequently migrates to left and behind anlage IV. These anlagen plus small anterior part of anlage I (as mentioned above) gradually differentiate into new cirri in the following pattern: the anterior part of anlage I generates the left frontal cirrus; anlage II forms the middle frontal cirrus and buccal cirrus; anlage III produces the right frontal cirrus and the parabuccal cirrus; anlage IV contributes to the frontoventral row in opisthe. In proter, anlage IV, along with anlage V, combine to form the frontoventral row (Figure 2P–R,T, Figure 3P–R,T, Figure 4A,C,E,G and Figure 5A,C,E,G).

Marginal rows anlagen appear earlier in opisthe than in proter. The anlage of inner right marginal row (RMA1) appears earlier than other marginal anlagen (Figure 2K and Figure 3K). It is apparent that in both proter and opisthe, two or three old cirri in the inner right marginal row undergo dedifferentiation and contribute to the formation of the RMA1 (Figure 2L and Figure 3L). In the proter, RA1 develops to the right of the parental inner right marginal row (Figure 2K,L,Q and Figure 3K,L,Q), while in the opisthe, RA1 is formed within the parental inner right marginal row (Figure 2R and Figure 3R). Other marginal anlagen develop intrakinetally within parental rows in each daughter cell (Figure 2T,U and Figure 3T,U). Subsequently, with the further proliferation of basal bodies, all marginal anlagen lengthen bidirectionally and generate new cirri to replace the old ones at the final stage of division (Figure 4A,C,E,G and Figure 5A,C,E,G).

##### Dorsal Ciliature

The dorsal kinety anlagen develop intrakinetally within the dorsal kineties 1–3 at two distinct levels corresponding to proter and opisthe (Figure 2M and Figure 3M). With the proliferation of basal bodies, these anlagen extend toward both ends, gradually replacing the old structures (Figure 2S, Figure 3S, Figure 4A,B and Figure 5A,B). As the macronuclear nodules gradually fuse into a singular mass, the left dorsal kinety anlage breaks into two parts (Figure 4A and Figure 5A). During the morphogenetic process, one caudal cirrus is formed at the posterior end of each dorsal kinety anlagen in both proter and opisthe (Figure 4D,F,H and Figure 5D,F,H).

##### Nuclear Apparatus

In the very early stage, a replication band is present in each macronuclear nodule (Figure 2A and Figure 3A). In middle dividers, the macronuclear nodules fuse together to form a single globular mass. Subsequently, this mass undergoes division three times before cytokinesis occurs, ensuring the generation of a sufficient number of macronuclear nodules for both proter and opisthe (Figure 2S,T, Figure 3S,T, Figure 4A–H and Figure 5B,D,F,H). The micronuclei divide mitotically (Figure 2T, Figure 3S,T, Figure 4A,C,F,H and Figure 5B,D,F,H).

Genus *Perisincirra* Jankowski, 1978

Improved diagnosis

Medium-sized, non-dorsomarginalian Deviatidae with frontal, buccal and caudal cirri present. Two or more marginal rows on each side of cell, with the cirri within these rows being very widely spaced; frontoventral row absent. Three dorsal kineties. The oral primordium originates apokinetally between right and left cirral field. Consistently four frontal-ventral cirral anlagen.

### 3.2. Perisincirra paucicirrata Foissner et al., 2002 [9]

#### 3.2.1. Improved Diagnosis

Size about 60–130 μm × 15–32 μm in vivo with an elongated ellipsoidal shape. Two macronuclear nodules and a single micronucleus located between. Single contractile vacuole at mid-body and close to left cell margin. Adoral zone occupying 10–25% of the body length in vivo and composed of 13–20 membranelles. One buccal, three frontal and two parabuccal cirri. Three or four left and consistently two right marginal rows. One caudal cirrus at the posterior end of each dorsal kinety.

#### 3.2.2. Voucher Slides

Three slides (registration number: LLJ2023050602/1–3) with protargol-stained morphostatic and dividing specimens have been deposited in the Laboratory of Systematic Taxonomy, Ocean University of China, China.

#### 3.2.3. Morphological Description of Qingdao Population

Cell size 75–125 μm × 15–30 μm (110 μm × 25 μm on average) in vivo (*n* = 10), slightly flattened dorsoventrally, with a length to width ratio ranging from 3.6 to 5.8 (4.7 on average) in vivo (*n* = 10); about 105 μm × 42 μm (75–135 μm × 32–58 μm) after protargol staining (Table 2). Cell flexible but not contractile, elongated ellipsoidal with margins slightly converging posteriorly, usually with the anterior part wider than the posterior part (Figure 6A,F–H). Two ellipsoidal macronuclear nodules, located to the left of the midline in the middle portion of the body (Figure 6A,C,E,M). The micronucleus (about 5 μm in diameter) usually near or at the posterior end of the anterior macronuclear nodule (Figure 6C,E,M). Contractile vacuole about 15 μm across, located at the equatorial level near the left cell margin (Figure 6A,H). Cytoplasm colourless, with numerous transparent colourless inclusions (2–11 μm), most being spherical and located in the middle of the cell, which renders a dark and opaque appearance (Figure 6F–H,J,L). Locomotion mainly by swimming or crawling on substrate; rotating about the longitudinal axis of the body when swimming.

Adoral zone occupying approximately 27% of body length in protargol preparations, composed of 18–20 (19 on average) membranelles roughly arranged in *Gonostomum* pattern. The bases of the largest membranelles about 5 μm wide and the cilia of distal membranelles up to about 13 µm in length in vivo (Figure 6A,B,D,G,I). Paroral and endoral membranes monokinetidal, mostly straight and parallel to each other (four out of 21 cells examined intersected spatially); paroral being about 3/4 of endoral in length and positioned more anteriorly (Figure 6B,D). Pharyngeal fibers conspicuous, extending obliquely backward and measuring approximately 20–35 μm long after protargol staining (Figure 6B,D). Constantly three slightly enlarged frontal cirri located near the anterior end of the cell, with the right one close to the distal end of adoral zone of membranelles; one buccal cirrus located in front of the anterior end of the endoral and to the right of the anterior end of paroral; two parabuccal cirri located behind the right frontal cirrus (Figure 6B,D,K). Invariably, three widely spaced left marginal rows, commencing near the proximal level of the adoral zone and comprising 12–16, 11–16, and 11–18 cirri from the inner row to the outer one, respectively (Figure 6A,B,D). Consistently two bipolar right marginal rows, with the inner row consisting of 18–23 cirri and the outer row of 17–22 cirri (Figure 6A,B,D). Marginal cirri remarkably thin and approximately 13 μm long in protargol-stained specimens (Figure 6D).

Three dorsal kineties composed of 3, 10, and 3 dikinetids from left to right, with the dikinetids in the left and right kineties being sparsely concentrated in both ends. Only the anterior kinetid bearing a bristle about 3 μm long in stained cells (Figure 6C,E). Three caudal cirri, with one located at the posterior end of each dorsal kinety (Figure 6E,L).

#### 3.2.4. Morphogenesis During Binary Fission

##### Stomatogenesis

Stomatogenesis commences with the de novo formation of the oral primordium (OP) of the opisthe in the postoral area between the inner right and left marginal rows (Figure 7A and Figure 8A). Subsequently, with the proliferation of basal bodies, the OP enlarges and differentiates into new adoral membranelles for the opisthe in its right anterior region (Figure 7B–E and Figure 8B–E). Concurrently, undulating membranes anlage (frontal-ventral cirral anlage I) is detached from the right side of the OP (Figure 7F,G and Figure 8F,G). In the later stages, the differentiation of membranelles is completed from anterior to posterior and the anterior end of the newly formed adoral zone curves to the right (Figure 7H–M and Figure 8H–M). The posterior majority of anlage I splits longitudinally to form the paroral and endoral membranes (Figure 7N,O,R and Figure 8N,O,R).

In the proter, the parental adoral zone of membranelles remains intact and is wholly inherited by the proter. The anlage of the left frontal cirrus is derived from the dedifferentiated anterior portion of the parental paroral (Figure 7K–O and Figure 8K–O), and the remaining portion of the parental paroral is resorbed entirely (Figure 7R and Figure 8R). The paroral and endoral anlage is formed from the reorganization of the parental endoral (Figure 7R, Figure 8R, Figure 9A and Figure 10A).

##### Ventral Ciliature

Frontal-ventral cirral anlagen II–IV for the opisthe appear earlier in opisthe than those for the proter. In the opisthe, two cirral anlagen (anlagen II and III) emerge to the right of anlage I (Figure 7H–L and Figure 8H–L). Then, anlage III splits to form anlagen III and IV (Figure 7M and Figure 8M). Anlage IV then migrates rightward and appears to the right rear of anlage III (Figure 7N,O,R and Figure 8N,O,R); for the proter, the parental buccal cirrus and anterior parabuccal cirrus dedifferentiate and contribute to the formation of anlagen II and III (Figure 7J–M and Figure 8J–M). Subsequently, the posterior parabuccal cirrus dedifferentiates into anlage IV (Figure 7N and Figure 8N), while the parental endoral begins to disintegrate. At this stage, four cirral anlagen are formed in both proter and opisthe (Figure 7N and Figure 8N), which gradually differentiate into new cirri in the following pattern: the anterior part of anlage I produces the left frontal cirrus; anlage II forms the middle frontal cirrus and buccal cirrus; anlage III gives rise to the right frontal cirrus and the anterior parabuccal cirrus; anlage IV generates the posterior parabuccal cirrus (Figure 9A–D,F–I and Figure 10A–D,F–I). Sometimes, a few extra cirri from anlagen III and IV will be resorbed before cytokinesis (Figure 9C,D,F,G and Figure 10C,D,F,G).

Marginal cirral anlagen originate intrakinetally within the parental marginal rows. It seems that a few cirri below the mid-body within each parental marginal row dedifferentiate to form anlage for the opisthe (Figure 7N and Figure 8N). Subsequently, a few cirri near the anterior end within each parental marginal row also dedifferentiate to form anlage for the proter (Figure 7R and Figure 8R). As basal bodies proliferate further, these anlagen lengthen bidirectionally and generate new cirri while the parental structures gradually disintegrate and are resorbed (Figure 9A–D,F–I and Figure 10A–D,F–I).

##### Dorsal Ciliature

The dorsal kineties anlagen develop intrakinetally within the parental kineties at two levels corresponding to proter and opisthe (Figure 7P and Figure 8P). With the proliferation of basal bodies, these anlagen extend toward both ends, gradually replacing the old structures (Figure 7Q, Figure 8Q, Figure 9E,J and Figure 10E,J). During the morphogenetic process, one caudal cirrus is formed at the posterior end of each dorsal kinety anlagen in both proter and opisthe (Figure 9E,J and Figure 10E,J).

##### Nuclear Apparatus

In very early dividers, DNA synthesis occurs in macronucleus, which is indicated by the presence of a replication band in each nodule (Figure 7F,H and Figure 8F,H). In middle dividers, the two macronuclear nodules merge to form a single globular mass. Subsequently, in late dividers, this mass undergoes division twice before cytokinesis to produce two macronuclear nodules for each daughter cell (Figure 9A,B,H,I and Figure 10A,B,H,I). The micronucleus splits into two mitotically and distributes to two daughter cells (Figure 9A,B,G,H and Figure 10A,B,G,H).

### 3.3. 18S rRNA Gene Sequences and Phylogenetic Analyses

The two newly obtained 18S rRNA gene sequences have been deposited in GenBank. The sequence of *Pseudosincirra binaria* sp. nov. (GenBank accession number PQ208515) is 1729 bp long with a G + C content of 44.77%; The sequence of Qingdao population of *Perisincirra paucicirrata* (GenBank accession number PQ208516) is 1730 bp long with a G + C content of 45.09%.

The phylogenetic trees constructed based on the 18S rRNA gene sequence data using Maximum Likelihood (ML) and Bayesian Inference (BI) analyses show mostly congruent results, although with some variations in support values between methods. Therefore, only the ML tree topology is presented here, with support values from both BI and ML analyses displayed (Figure 11).

In the phylogenetic trees, members of the family Deviatidae with known sequences cluster together with low to moderate support (90% ML, 0.80 BI). Within this cluster, there are two clades. One clade includes species such as *Perisincirra paucicirrata*, *Deviata bacilliformis* (Gelei, 1954) Eigner, 1995 [34], *D. multilineae* Zhang et al., 2022 [37], *D. brasiliensis* Siqueira-Castro et al., 2009 [50], and *D. abbrevescens* Eigner, 1995 [34] (75% ML, 0.83 BI). The other clade consists of *Deviata rositae* Küppers et al., 2007 [51], *D. parabacilliformis* Li et al., 2014 [52], *Heterodeviata sinica* Song et al., 2023 [28], *H. nantongensis* Liao et al., 2024 [29], *P. longicirrata* (Foissner et al., 2002) Gao et al., 2021 [24], and the new species investigated in present study (59% ML), which are grouped into three parallel subclades including two *Deviata* species, two *Heterodeviata* species and two *Pseudosincirra* species, respectively in BI analysis. The sister relationship between *P. binaria* sp. nov. and *P. longicirrata* is highly supported (96% ML), and the closer relationship between two Chinese populations of *P. paucicirrata* is fully supported (100% ML, 1.00 BI). In terms of sequence, the new species differs from *P. longicirrata* by 26 nucleotides (corresponding to 98.3% sequence similarity). Compared to other deviatids, the new species shows differences of 23–33 nucleotides (corresponding to 97.8–98.5% similarities) from seven species of *Deviata*, 25 nucleotides (corresponding to 98.3% similarity) from two populations of *P. paucicirrata*, 34 nucleotides (corresponding to 97.7% similarity) from *H. sinica* and *H. nantongensis* (Table 3).

## 4. Discussion

### 4.1. Identification of Pseudosincirra binaria sp. nov.

In view of several longitudinal cirral rows dividing intrakinetally, the absence of transverse cirri, the oral primordium originating apokinetally between the right and left cirral field and four or five frontal-ventral cirral anlagen, *Pseudosincirra binaria* sp. nov. should be assigned to the family Deviatidae. Compared to other extant genera within the family, the new species can be easily distinguished from *Deviata* Eigner, 1995 and *Idiodeviata* Foissner, 2016 by the presence (vs. absence) of caudal cirri [7,34], from *Perisincirra* Jankowski, 1978 by the presence (vs. absence) of frontoventral row [35], from *Notodeviata* Foissner, 2016 by the number of bipolar dorsal kineties (three vs. two in *Notodeviata*) and the absence of the peculiar morphogenetic feature, i.e., all new cirri of anlage IV and most cirri of anlagen II and III are resorbed in late and very late dividers [7] and from *Heterodeviata* Song et al., 2023 by the number of bipolar dorsal kineties (three vs. one in *Heterodeviata*) and the absence of dorsomarginal kinety (vs. presence in the latter) [28]. The current species undoubtedly belongs to the monotypic genus *Pseudosincirra*. Our species can be clearly separated from *Pseudosincirra longicirrata* by the absence (vs. presence) of dorsomarginal kinety, the bipartite left dorsal kinety (presence vs. absence) and the number of macronuclear nodules (four vs. two in *P. longicirrata*) [24]. Thus, a new species has to be proposed.

### 4.2. Morphogenetic Comparison

Considering the formative modes of frontal-ventral anlagen and dorsal kineties, we must compare *Pseudosincirra binaria* sp. nov. with similar species that possess three clearly differentiated frontal cirri within the Deviatidae. Till now, the morphogenesis of ten deviatids species has been investigated in detail, i.e., *Deviata abbrevescens*, *D. baciliformis*, *D. brasiliensis*, *D. parabacilliformis*, *Idiodeviata venezuelensis*, *Notodeviata halophila*, *Perisincirra paucicirrata*, *Pseudosincirra longicirrata*, *Heterodeviata sinica* and *Heterodeviata nantongensis* [7,24,28,29,34,51,52,53]. Four *Deviata* species differ from our species by the number of frontal-ventral cirral anlagen (six vs. four or five in our species) and the absence (vs. presence) of caudal cirri formed at posterior ends of dorsal kineties anlagen [34,51,52,53]. *Idiodeviata venezuelensis* can be distinguished from the new species by the dorsal kinety anlage that does not produce caudal cirrus at its rear end and the presence of dorsomarginal kinety anlage [7]. Though *Notodeviata halophila* share the absence of dorsomarginal kinety anlage with *Pseudosincirra binaria* sp. nov., it is clearly distinct from the latter by its peculiar morphogenetic feature, i.e., all new cirri of anlage IV and most cirri of anlagen II and III are resorbed in late and very late dividers [7]. Both *Perisincirra paucicirrata* and *Pseudosincirra binaria* sp. nov. share the absence of dorsomarginal kinety anlage, but the former does not develop frontoventral row. The new species investigated here differs from *Heterodeviata sinica*, *H. nantongensis* and *Pseudosincirra longicirrata* by the absence of dorsomarginal kinety anlage. Additionally, *Heterodeviata sinica* differs from our species by the origin of frontoventral cirral row (from single anlage in both proter and opisthe in *H. sinica* vs. from two anlagen in proter and single anlage in opisthe). Moreover, *Pseudosincirra binaria* sp. nov. differs from *Heterodeviata sinica* and *H. nantongensis* by more dorsal kineties anlagen (three vs. two) and the dividing of the left dorsal kinety anlage in late stages of morphogenesis [24,28,29].

While in Kahllidae, the parental frontoventral or/and marginal rows or/and dorsal kineties of most species are preserved, i.e., *Kahaliella simplex* (Horváth, 1934) Berger, 2011 (parental left marginal rows retained), *Neogeneia hortualis* Eigner, 1995 (parental marginal cirri retained), and *Parakahliella macrostoma* (Foissner, 1982) Berger et al., 1985 (parental dorsal kineties retained), which differs from *Pseudosincirra binaria* sp. nov. (no old structures kept in interphase cell) [20,34]. However, *Afrokahliella paramacrostoma* Li et al., 2021 and *Fragmocirrus espeletiae* Foissner, 2000 have no parental structures retained, but they can be distinguished from *Pseudosincirra binaria* sp. nov. by the presence (vs. absence in our new species) of dorsomarginal kinety and the generation of *Urosomoida*-patterned dorsal ciliature [54,55].

### 4.3. Identification of Perisincirra paucicirrata and Comparison with Congeners

In view of the absence of frontoventral row and dorsomarginal kinety as well transverse cirri, and the presence of four frontal ventral cirral anlagen, and two right and three left marginal cirral rows, the Qingdao population ought to be assigned to the genus *Perisincirra*. So far, there are only two *Perisincirra* species, namely, *Perisincirra kahli* (type species) and *Perisincirra paucicirrata* (Table 4) [9,20,36]. *Perisincirra kahli* differs from our population by distinctly higher ratio of length to width in vivo (10–15:1 vs. 4.7:1) and fewer left marginal rows (2 vs. 3) [20]. Morphologically, our population correspond well with the population described by Li et al. (2013), such as the body shape, the presence of two parabuccal cirri, the numbers of marginal rows, dorsal kineties, and caudal cirri [36]. Therefore, our population can be identified as *Perisincirra paucicirrata.* For comparative purposes, details of main morphological features of *Perisincirra paucicirrata* and its congener *P. kahli* (Groliere 1975) Jankowski 1978 are given in Table 4.

Since the establishment of the genus *Perisincirra*, only the morphogenesis of *P. paucirirrata* has been studied once [36]. The main morphogenetic features of our population correspond well with those described by Li et al., (2013) [36]: (i) four frontal-ventral cirral anlagen are present; (ii) in the opisthe, anlage I, II, and III derived from the oral primordium, anlage IV derived from the splitting of anlage III; (iii) in the proter, anlage I, III and IV originate from the anterior portion of the parental paroral, the parental anterior parabuccal cirrus and the parental posterior parabuccal cirrus; (iv) anlage III splits longitudinally to form anlagen III and IV in the proter; (v) the marginal rows and the dorsal kineties develop intrakinetally; (vi) no parental structures are retained after division, except for the parental adoral zone of membranelles inherited by the proter; and (vii) complete fusion of macronuclear nodules into a single mass, and the micronuclei divide mitotically [36]. However, in terms of the proter, in our population, parental buccal cirrus dedifferentiates to form anlage II, in the population described by Li et al., (2013) [36], parental buccal cirrus is completely resorbed and anlage II originates de novo. Berger (2011) [20] speculated that the species in the genus *Perisincirra* may have three frontal-ventral anlagen, but could not define the inner right row in either *Perisincirra kahli* or *P. paucicirrata* is marginal row or frontoventral row. Based on the investigation of *Perisincirra paucicirrata*, Li et al., (2013) judged that the inner cirral row right of midline in *Perisincirra kahli* is probably the right marginal row, which is consistent with the findings of the present study [36].

### 4.4. Phylogenetic Analyses

*Pseudosincirra binaria* sp. nov. falls in a clade that includes six *Deviata* species (seven popuplations), *Perisincirra paucicirrata* (two populations), *Pseudosincirra longicirrata*, *Heterodeviata sinica* and *H. nantongensis*. The close relationship between these eleven species is supported by the shared following characteristics: the adoral zone of membranelles roughly in *Gonostomum* pattern; at least one left and one right marginal row, fine cirri in the ventral and marginal rows and cirri within all rows relatively widely spaced; three frontal cirri; buccal, and parabuccal cirrus/cirri present; frontoterminal, postoral ventral, pretransverse ventral, and transverse cirri lacking; typical parental (old) cirral rows resorbed after cytokinesis. The presence or absence of two traits such as dorsomarginal kineties and caudal cirri as well the numbers of dorsal kineties and frontoventral rows vary significantly in these Deviatidae species. Foissner (2016) proposed that the dorsomarginal kineties might have evolved several times independently and inferred that Deviatidae is possibly sister to the non-dorsomarginalian family Kahliellidae Tuffrau, 1979, which was not confirmed by the present study and recent studies [24,28,29,52,53]. Deviatidae show a closer relationship with Dorsomarginalia and *Strongylidium–Hemiamphisiella–Pseudouroleptus* rather than Kahliellidae. Three distinct genera within Kahliellidae (*Kahliella*, *Parakahliella* and *Engelmanniella*) branch separately from Deviatidae, which is consistent with the finding of the recent work [24,28,29,37].

In the present study, the well-grouping of all deviatid species with SSU rRNA gene sequence data available supports the rationality of the established family Deviatidae by Foissner [7] and the monophyly of the family, which is consistent with previous studies [24,28,29,37]. The monophyly of Deviatidae is also supported by the AU test (*p* = 0.941). Furthermore, six *Deviata* species group with members of genera *Perisincirra*, *Pseudosincirra*, *Heterodeviata* rather group in a single clade, which demonstrates that *Deviata* is non-monophyletic. However, the monophyly of *Deviata* was not rejected by the AU test (*p* = 0.659). The sister relationship between *Pseudosincirra binaria* sp. nov. and *Pseudosincirra longicirrata* conforms to their high level of similarities as regards morphology and morphogenetic pattern [24 and comparison above]. However, their identities as distinct species are corroborated by sequence divergences and discrepancies in other morphological and morphogenetic traits as discussed above [24]. The well-clustering of two population of *Perisincirra paucicirrata* is also consistent with their morphological and morphogenetic similarities [36]. Gao et al. [24] suggested that *Perisincirra paucicirrata* should be assigned to Deviatidae, which is supported by the present study. However, since molecular data are not available for the type species of *Kahliella* and *Perisincirra* (*K. acrobates* and *P. kahli* respectively), conclusion on the familial assignment of both genera seem to be premature.

## Figures and Tables

**Figure 1 microorganisms-12-02013-f001:**
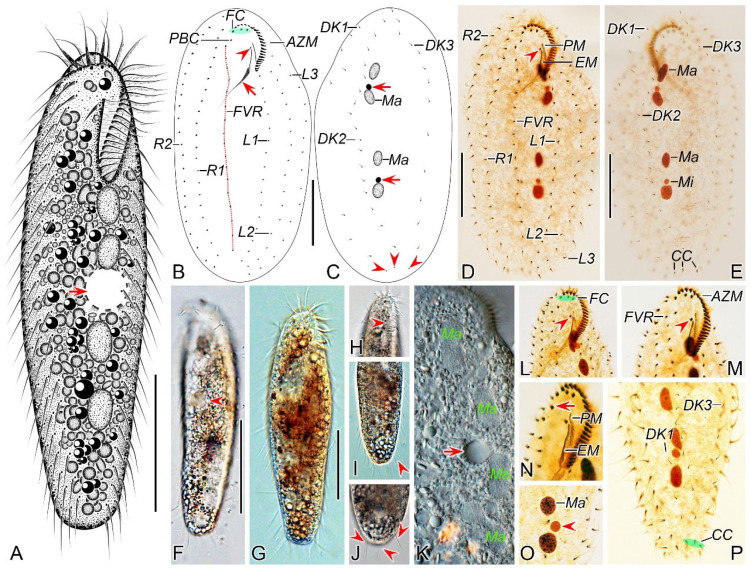
Morphology of *Pseudosincirra binaria* sp. nov. from life (**A**,**F**–**K**) and after protargol staining (**B**–**E**,**L**–**P**). (**A**) Ventral view of a representative individual, arrow indicates the contractile vacuole. (**B**,**C**) Ventral (**B**) and dorsal (**C**) views of the holotype, showing the infraciliature and nuclear apparatus, arrowhead and arrow in (**B**) mark the buccal cirrus and pharyngeal fibers, respectively, arrows and arrowheads in (**C**) denote micronuclei and caudal cirri, separately. (**D**,**E**) Ventral (**D**) and dorsal (**E**) views of the same specimen, to show the general infraciliature and nuclear apparatus, arrowhead showing buccal cirrus. (**F**) Dorsal view, showing the contractile vacuole (arrowhead). (**G**) Ventral view of typical well-nourished cultured individual. (**H**) Detail of the anterior end of body, arrowhead denotes the adoral membranelles. (**I**) Detail of the posterior portion of cell, arrowhead indicates the marginal cirri. (**J**) Detail of the posterior end of body, arrowheads show caudal cirri. (**K**) Showing the cytoplasmic granules, macronuclear nodules and contractile vacuole (arrow). (**L**,**M**) Ventral views of the anterior portion of two cells, showing oral apparatus, frontal cirri and the buccal cirrus (arrowheads), and part of frontoventral row. (**N**) Ventral view of the anterior portion of cell, showing paroral and endoral membranes, and parabuccal cirrus (arrow). (**O**) Two macronuclear nodules and micronucleus in between (arrowhead). (**P**) Detail of the posterior portion of cell, showing dorsal kineties and caudal cirri. AZM, adoral zone of membranelles; CC, caudal cirri; DK1, 2, 3, dorsal kineties 1, 2, 3; EM, endoral membrane; FC, frontal cirri; FVR, frontoventral row; L1, 2, 3, the inner, middle and outer left marginal row; Ma, macronuclear nodules; Mi, micronuclei; PBC, parabuccal cirrus; PM, paroral membrane; R1, 2, the inner and outer right marginal row. Scale bars = 50 μm.

**Figure 2 microorganisms-12-02013-f002:**
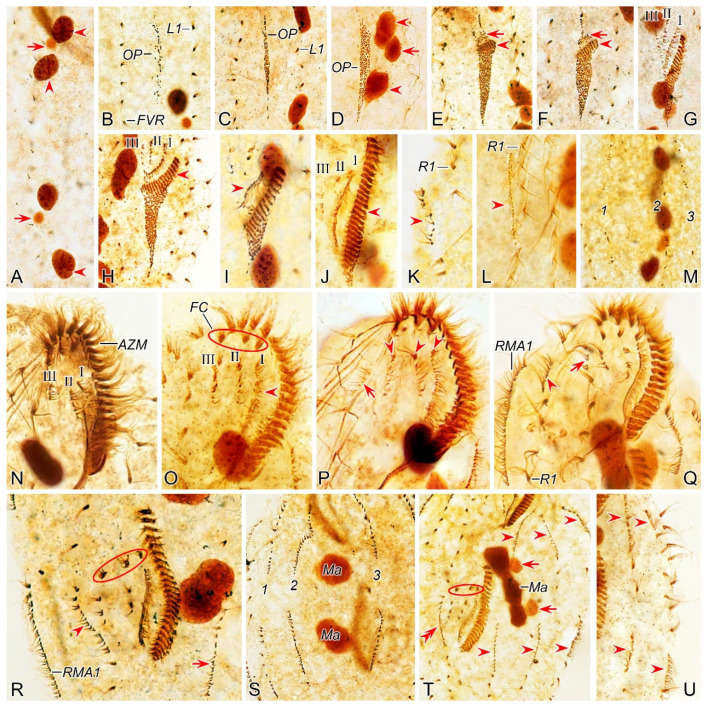
Morphogenetic photomicrographs of *Pseudosincirra binaria* sp. nov. after protargol staining. (**A**) Detail of early divider, arrowheads indicate the replication bands and arrows denote the micronuclei. (**B**) Ventral view of a very early divider, showing the newly formed oral primordium between frontoventral row and the inner left marginal row. (**C**,**D**) With the proliferation of basal bodies, the oral primordium of the opisthe lengthens; arrowheads and arrow in (**D**) indicate macronuclear nodules and the micronucleus, respectively. (**E**,**F**) Ventral views of early dividers, arrowheads denote several newly formed adoral zone of membranes and arrows point to undulating membranes anlage, in this stage, streaks are formed anteriorly to the right of the oral primordium. (**G**,**H**) Ventral views of early dividers, arrowhead in (**H**) demonstrates incompletely formed adoral zone of membranes, in this stage, anlagen I–III of the opisthe are formed. (**I**,**J**) Ventral views of early dividers, arrowhead in (**I**) showing the posterior majority of anlage I splits longitudinally to form paroral and endoral membranes and arrowhead in (**J**) marks the almost completely formed adoral zone of membranes. (**K**,**L**) Ventral views of early dividers, showing the anlage for the inner right marginal row of the opisthe (arrowheads) formed de novo to the right of the parental row. (**M**) Dorsal view of an early divider, showing dorsal kineties anlagen 1–3 (number). (**N**) Ventral view of the anterior portion of an early divider, showing the anterior portion of the parental paroral dedifferentiating to anlage I, buccal cirrus and parabuccal cirrus dedifferentiating to anlage II and anlage III, respectively. (**O**) Ventral view of the anterior portion of an early divider, showing the parental frontal cirri (ellipse) and frontal ventral cirral anlagen I–III of the proter, arrowhead shows the parental undulating membranes begin to disintegrate. (**P**) Ventral view of the anterior portion of an early divider, arrowheads mark newly formed frontal cirri and arrow points to frontal ventral cirral anlage IV of the proter. (**Q**) Ventral view of the anterior portion of an early divider, showing the anlage of inner right marginal row develops to the right of the parental inner right marginal row, arrowhead indicates anlage V of the proter appears within the parental inner right marginal row and arrow denotes frontal ventral cirral anlage IV. (**R**,**S**) Ventral (**R**) and dorsal (**S**) views of the same early divider, arrow and arrowhead in (**R**) indicate the left marginal row anlage I and intrakinetally formed frontal ventral cirral anlage IV of the opisthe, respectively, ellipse encircles the newly formed frontal cirri of the opisthe, numbers in (**S**) demonstrate dorsal kineties anlagen 1–3 formed within old kineties. (**T**) Ventral view of an early-middle divider, arrowheads denote the intrakinetally formed marginal row anlagen, arrows mark the micronuclei, double arrowhead indicates frontal oventral cirral anlage IV of the opisthe and ellipse encircles the newly formed frontal cirri of the opisthe, in this stage, the macronuclear nodules begin to fuse into a single globular mass. (**U**) Ventral view of an early divider, arrowheads point to left marginal row anlagen, in this stage, only anlagen for the inner and middle left marginal rows appear. 1–3, dorsal kineties anlagen 1–3; AZM, adoral zone of membranelles; FC, frontal cirri; FVR, frontoventral row; L1, inner left marginal row; Ma, macronuclear nodules; OP, oral primordium; R1, inner right marginal row; RMA1, anlage for the inner right marginal row; I–III, frontal ventral cirral anlagen I–III. Scale bars = 30 μm.

**Figure 3 microorganisms-12-02013-f003:**
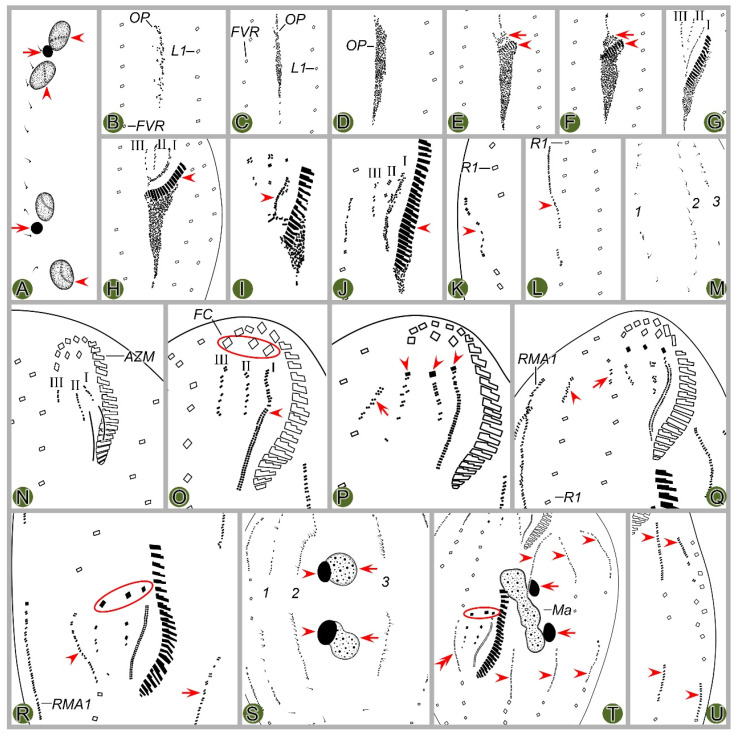
Morphogenesis of *Pseudosincirra binaria* sp. nov. after protargol staining. (**A**) Detail of early dividers, arrowheads indicate the replication bands and arrows denote the micronuclei. (**B**) Ventral view of a very early divider, showing the newly formed oral primordium between the frontoventral row and the inner left marginal row. (**C**,**D**) With the proliferation of basal bodies, the oral primordium of the opisthe lengthens. (**E**,**F**) Ventral views of early divider, arrowheads denote several newly formed adoral zone of membranes and arrows point to undulating membranes anlage, in this stage, streaks are formed anteriorly to the right of the oral primordium. (**G**,**H**) Ventral views of early divider, arrowhead in (**H**) demonstrates incompletely formed adoral zone of membranes, in this stage, anlagen I–III of the opisthe are formed. (**I**,**J**) Ventral views of early divider, arrowhead in (**I**) showing the posterior majority of anlage I splits longitudinally to form paroral and endoral membranes and arrowhead in (**J**) marks the almost completely formed adoral zone of membranes. (**K**,**L**) Ventral views of early divider, showing the anlage for the inner right marginal row of the opisthe (arrowheads) formed de novo to the right of the parental row. (**M**) Dorsal view of an early divider, showing dorsal kinety anlagen 1–3 (number). (**N**) Ventral view of the anterior portion of an early divider, showing the anterior portion of the parental paroral dedifferentiating to anlage I, buccal cirrus and parabuccal cirrus dedifferentiating to anlage II and anlage III, respectively. (**O**) Ventral view of the anterior portion of an early divider, showing the parental frontal cirri (ellipse) and frontal ventral cirral anlagen I–III of the proter, arrowhead shows the parental undulating membranes begin to disintegrate. (**P**) Ventral view of the anterior portion of an early divider, arrowheads mark newly formed frontal cirri and arrow points to frontal ventral cirral anlage IV of the proter. (**Q**) Ventral view of the anterior portion of an early divider, showing the anlage of inner right marginal row develops to the right of the parental inner right marginal row, arrowhead indicates anlage V of the proter appears within the parental inner right marginal row and arrow denotes frontal ventral cirral anlage IV. (**R**,**S**) Ventral (**R**) and dorsal (**S**) views of the same early divider, arrow and arrowhead in (**R**) indicate the left marginal row anlage I and intrakinetally formed frontal ventral cirral anlage IV of the opisthe, respectively, ellipse encircles the newly formed frontal cirri of the opisthe, numbers in (**S**) demonstrate dorsal kineties anlagen 1–3 formed within old kineties. (**T**) Ventral view of an early-middle divider, arrowheads denote the intrakinetally formed marginal cirral anlagen, arrows mark the micronuclei, double arrowhead indicates frontal ventral cirral anlage IV of the opisthe and ellipse encircles the newly formed frontal cirri of the opisthe, in this stage, the macronuclear nodules begin to fuse into a single globular mass. (**U**) Ventral view of an early divider, arrowheads point to left marginal row anlagen, in this stage, only anlagen for the inner and middle left marginal rows appear. 1–3, dorsal kinety anlagen 1–3; AZM, adoral zone of membranelles; FC, frontal cirri; FVR, frontoventral row; L1, inner left marginal row; Ma, macronuclear nodules; OP, oral primordium; R1, inner right marginal row; RMA1, anlage for the inner right marginal row; I–III, frontal ventral cirral anlagen I–III. Scale bars = 30 μm.

**Figure 4 microorganisms-12-02013-f004:**
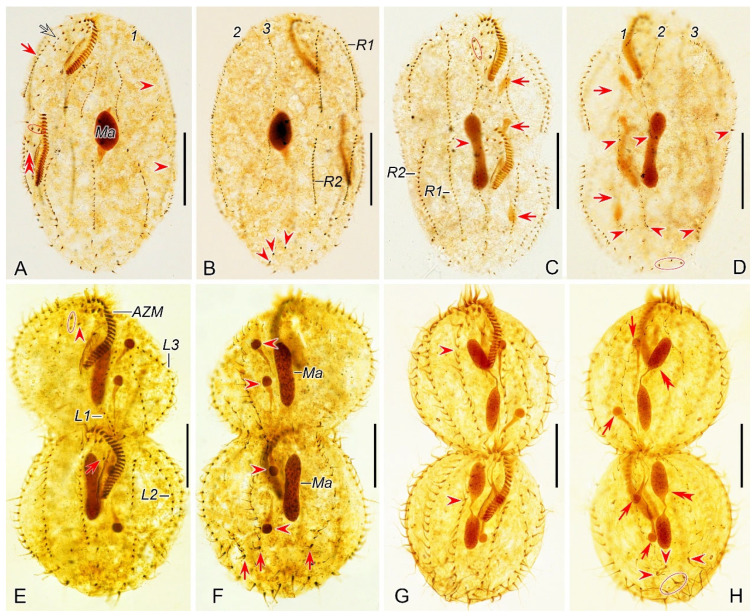
Morphogenetic photomicrographs of *Pseudosincirra binaria* sp. nov. after protargol staining. (**A**,**B**) Ventral (**A**) and dorsal (**B**) views of the same middle divider, solid arrow marks the anlage V of the proter develops into new cirri posteriad, hollow arrow denotes new cirri from the anlage IV of the proter, double-arrowhead indicates the newly formed parabuccal cirri of the opisthe, ellipse encircles the newly formed frontal cirri of the opisthe, arrowheads in (**A**,**B**) show the conspicuous gap in the middle part of inner dorsal kinety and the parental caudal cirri, respectively, in this stage, the macronuclear nodules fuse into a single mass. (**C**,**D**) Ventral (**C**) and dorsal (**D**) views of the same middle-late divider, arrows in (**C**) denote the micronuclei, arrowhead indicates the fuse macronuclear mass begins to divide, arrowheads in (**D**) demonstrate the newly formed caudal cirri of the proter and opisthe, arrows indicate the conspicuous gap in the middle part of the left dorsal kinety, ellipses encircle the cirri developed from the anlage IV in (**C**) and the parental caudal cirri in (**D**). (**E**,**F**) Ventral (**E**) and dorsal (**F**) views of the same late divider, arrow and arrowhead in (**E**) indicate the newly formed buccal cirrus of the opisthe and the parabuccal cirrus of the proter, ellipse encircles the cirri developed from the anlage IV, arrows and arrowheads in (**F**) point to the new caudal cirri and micronuclei, respectively, in this stage, the macronucleus complete its first division. (**G**,**H**) Ventral (**G**) and dorsal (**H**) views of the same late divider, arrowheads in (**G**) indicate frontoventral row formed from cirri that developed from the anlagen IV and V in proter, and from anlage IV in opisthe, arrows and arrowheads in (**H**) denote micronuclei and caudal cirri, respectively, double-arrowheads indicate macronuclear nodules, ellipse encircles the old caudal cirri, in this stage, the macronucleus completes its second division. 1–3, dorsal kineties anlagen 1–3; AZM, adoral zone of membranelles; L1, 2, 3, the inner, middle and outer left marginal rows; Ma, fused macronuclear mass; R1, 2, the inner and outer right marginal rows. Scale bars = 30 μm.

**Figure 5 microorganisms-12-02013-f005:**
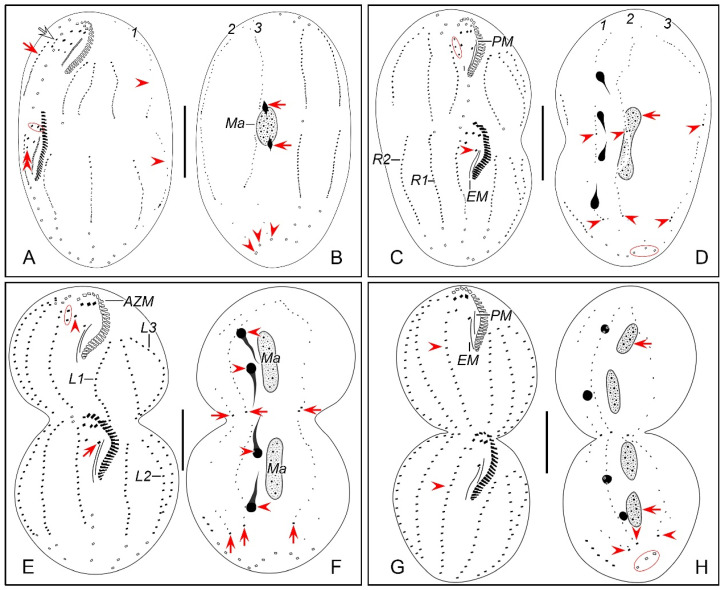
Middle to late morphogenetic stages of *Pseudosincirra binaria* sp. nov. after protargol staining. (**A**,**B**) Ventral (**A**) and dorsal (**B**) views of the same middle divider, solid arrow in (**A**) marks the anlage V of proter develops into new cirri posteriad, hollow arrow denotes new cirri from the anlage IV of the proter, double-arrowhead indicates the newly formed parabuccal cirri of the opisthe, ellipse encircles the newly formed frontal cirri of the opisthe, arrowheads in (**A**,**B**) show the conspicuous gap in the middle part of inner dorsal kinety and the parental caudal cirri, respectively, arrows in (**B**) indicate the micronuclei, in this stage, the macronuclear nodules fuse into a single mass. (**C**,**D**) Ventral (**C**) and dorsal (**D**) views of the same middle-late divider, arrowhead in (**C**) indicates the buccal cirrus of the opisthe, arrowheads in (**D**) demonstrate the newly formed caudal cirri of the proter and opisthe, arrow indicates the fuse macronuclear mass begins to divide, ellipses encircle the cirri developed from the anlage IV in (**C**) and the parental caudal cirri in (**D**). (**E**,**F**) Ventral (**E**) and dorsal (**F**) views of the same late divider, arrow and arrowhead in (**E**) indicate the newly formed buccal cirrus of the opisthe and the parabuccal cirrus of the proter, ellipse encircles the cirri developed from the anlage IV, arrows and arrowheads in (**F**) point to the new caudal cirri and micronuclei, respectively, in this stage, the macronucleus complete its first division. (**G**,**H**) Ventral (**G**) and dorsal (**H**) views of the same late divider, arrowheads in (**G**) indicate frontoventral row formed from cirri that developed from the anlagen IV and V in proter, and from anlage IV in opisthe, arrows and arrowheads in (**H**) denote macronuclear nodules and caudal cirri, respectively, ellipse encircles the old caudal cirri, in this stage, the macronucleus complete its second division. 1–3, dorsal kineties anlagen 1–3; AZM, adoral zone of membranelles; EM, endoral membrane; L1–3, the inner, middle and outer left marginal rows; Ma, fused macronuclear mass; PM, paroral membrane; R1, 2, the inner and outer right marginal rows. Scale bars = 30 μm.

**Figure 6 microorganisms-12-02013-f006:**
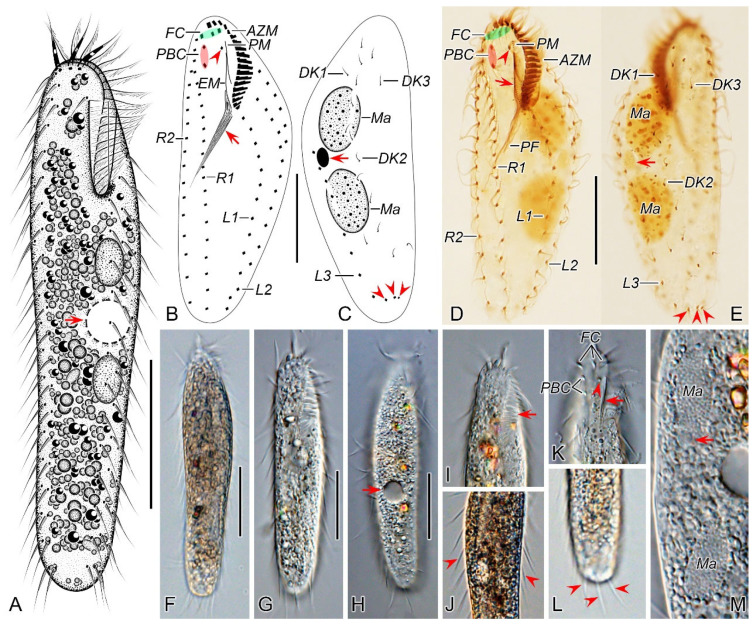
Morphology of *Perisincirra paucicirrata* from life (**A**,**F**–**M**) and after protargol staining (**B**–**E**). (**A**) Ventral view of a representative individual, arrow indicates contractile vacuole. (**B**,**C**) Ventral (**B**) and dorsal (**C**) views of the same cell, showing infraciliature and nuclear apparatus, arrowhead and arrow in (**B**) mark buccal cirrus and pharyngeal fibers, respectively, arrow and arrowheads in (**C**) denote micronucleus and caudal cirri, separately. (**D**,**E**) Ventral (**D**) and dorsal (**E**) views of the same specimen as in (**B**,**C**), to show the general morphology, arrow and arrowhead in (**D**) separately indicate endoral membrane and the buccal cirrus, arrow and arrowheads in (**E**) mark micronucleus and caudal cirri, respectively. (**F**,**G**) Ventral views of well-nourished cultured individuals showing shape variation. (**H**) Dorsal view, showing contractile vacuole (arrow). (**I**) Detail of the anterior portion of body, arrow marks adoral membranelles. (**J**) Detail of the middle part of cell, arrowheads indicate marginal cirri. (**K**) Detail of the anterior part of body, showing frontal cirri, parabuccal cirri, buccal lip (arrow) and buccal cirrus (arrowhead). (**L**) Detail of the posterior end of cell, arrowheads point to caudal cirri. (**M**) Showing the cytoplasm and nuclear apparatus, arrow indicates micronucleus (close to the anterior macronuclear nodule). AZM, adoral zone of membranelles; DK1–3, dorsal kineties 1–3; EM, endoral membrane; FC, frontal cirri; L1–3, inner, middle and outer left marginal row; Ma, macronuclear nodules; PBC, parabuccal cirri; PF, pharyngeal fibers; PM, paroral membrane; R1, 2, inner and outer right marginal row. Scale bars = 30 μm.

**Figure 7 microorganisms-12-02013-f007:**
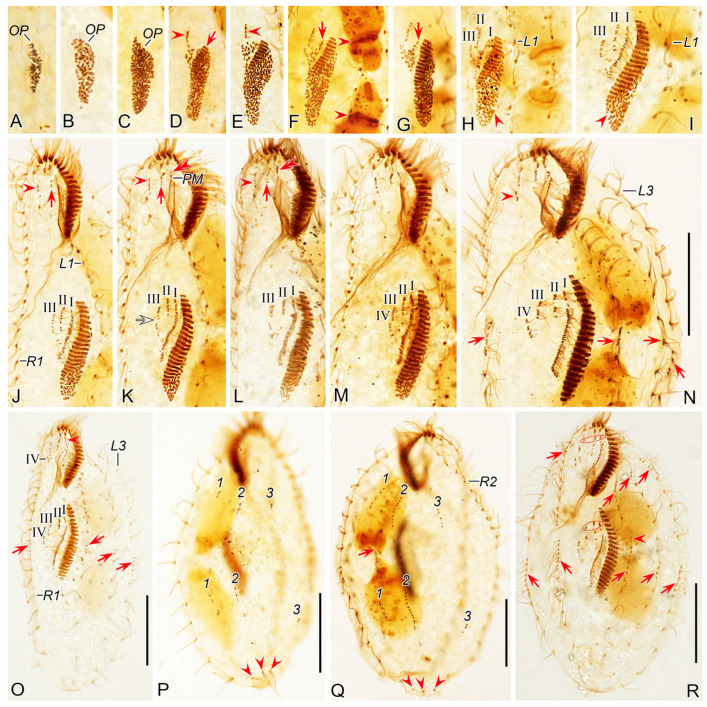
Morphogenesis photomicrographs of *Perisincirra paucicirrata* after protargol staining. (**A**) Ventral view of a very early divider to show the newly formed oral primordium between right and left cirral rows in opisthe. (**B**,**C**) With the proliferation of basal bodies, the oral primordium of the opisthe lengthens. (**D**,**E**) Ventral views of early dividers, arrow in (**D**) denotes the newly formed adoral membranes at the anterior end of OP and arrowheads in (**D**,**E**) point to the anlage II formed to the right anterior of the OP. (**F**,**G**) Ventral views of early dividers, arrowheads in (**F**) indicate replication bands and arrows in (**F**,**G**) denote undulating membranes anlage (anlage I). (**H**,**I**) Ventral views of early dividers, arrowheads mark incompletely formed adoral zone of membranes, in this stage, anlagen I–III of the opisthe are formed from the OP. (**J**) Ventral view of an early divider, arrow and arrowhead mark the dedifferentiation of buccal cirrus and anterior parabuccal cirrus. (**K**–**M**) Ventral views of early dividers, double-arrowhead indicates the anterior part of paroral disorganise and develop into anlage I of the proter, arrow and arrowhead denote anlagen II and III of the proter, respectively, hollow arrow demonstrates anlage III begins to split into anlagen III and IV. (**N**) Ventral view of early divider, showing anlage IV of the opisthe migrates rightward, arrowhead in (**N**) marks the dedifferentiation of posterior parabuccal cirrus, arrows point to marginal rows anlagen, in this stage cirral anlagen begin to develop into new cirri. (**O**) Ventral view of an early divider, arrowhead denotes the new left frontal cirrus and arrows indicate marginal rows anlagen, in this stage, anlagen I–IV appear in both proter and opisthe. (**P**) Dorsal view of an early divider, numbers mark dorsal kineties anlagen 1–3 forming within each dorsal kinety and arrowheads demonstrate the parental caudal cirri. (**Q**) Dorsal view of an early to middle divider, numbers mark dorsal kineties anlagen 1–3 formed within old structures, arrow denotes the dividing micronucleus and arrowheads point to the old caudal cirri. (**R**) Ventral view of an early to middle divider, arrows and arrowhead indicate marginal rows anlagen developing into new cirri posteriad in both proter and opisthe and the dividing micronucleus, respectively, ellipses encircle the newly formed frontal cirri of the proter and opisthe. 1–3, dorsal kineties anlagen 1–3; L1, 3, inner and outer left marginal rows; OP, oral primordium; PM, paroral membrane; R1, 2, inner and outer right marginal row 1, 2; I–IV, frontal ventral cirral anlagen I–IV. Scale bars = 30 μm.

**Figure 8 microorganisms-12-02013-f008:**
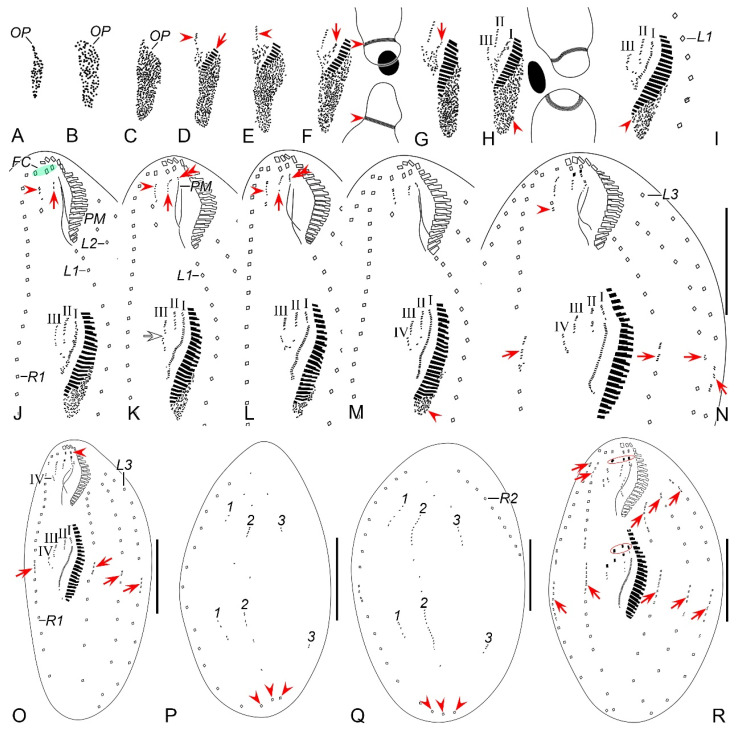
Morphogenesis of *Perisincirra paucicirrata* after protargol staining. (**A**) Ventral view of a very early divider to show the newly formed oral primordium between right and left cirral rows in opisthe. (**B**,**C**) With the proliferation of basal bodies, the oral primordium of the opisthe lengthens. (**D**,**E**) Ventral views of early dividers, arrow in (**D**) denotes the newly formed adoral membranes at the anterior end of OP and arrowheads in (**D**,**E**) point to anlage II formed to the right anterior of the OP. (**F**,**G**) Ventral views of early dividers, arrowheads in (**F**) indicate replication bands and arrows in (**F**,**G**) denote undulating membranes anlage (anlage I). (**H**,**I**) Ventral views of early dividers, arrowheads mark incompletely formed adoral zone of membranes, in this stage, anlagen I–III of the opisthe are formed from the OP. (**J**) Ventral view of an early divider, arrow and arrowhead mark the dedifferentiation of buccal cirrus and anterior parabuccal cirrus. (**K**–**M**) Ventral views of early dividers, double-arrowhead indicates the anterior part of paroral disorganise and develop into anlage I of the proter, arrow and arrowhead denote anlagen II and III of the proter, respectively, hollow arrow demonstrates anlage III begins to split into anlagen III and IV, arrowhead in (**M**) marks the incompletely formed adoral zone of membranes. (**N**) Ventral views of early divider, showing anlage IV of the opisthe migrates rightward, arrowhead in (**N**) marks the dedifferentiation of posterior parabuccal cirrus, arrows point to marginal rows anlagen, in this stage, cirral anlagen begin to develop into new cirri. (**O**) Ventral view of an early divider, arrowhead denotes the new left frontal cirrus and arrows indicate marginal rows anlagen, in this stage, anlagen I–IV appear in both proter and opisthe. (**P**) Dorsal view of an early divider, numbers mark dorsal kineties anlagen 1–3 forming within each dorsal kinety and arrowheads demonstrate the parental caudal cirri. (**Q**) Dorsal view of an early to middle divider, numbers mark dorsal kineties anlagen 1–3 formed within old structures, arrowheads point to the old caudal cirri. (**R**) Ventral view of an early to middle divider, arrows indicate marginal rows anlagen developing into new cirri posteriad in both proter and opisthe, ellipses encircle the newly formed frontal cirri of the proter and opisthe. 1–3, dorsal kineties anlagen 1–3; L1–3, inner, middle and outer left marginal rows; OP, oral primordium; PM, paroral membrane; R1, 2, inner and outer right marginal row 1, 2; I–IV, frontal ventral cirral anlagen I–IV. Scale bars = 30 μm.

**Figure 9 microorganisms-12-02013-f009:**
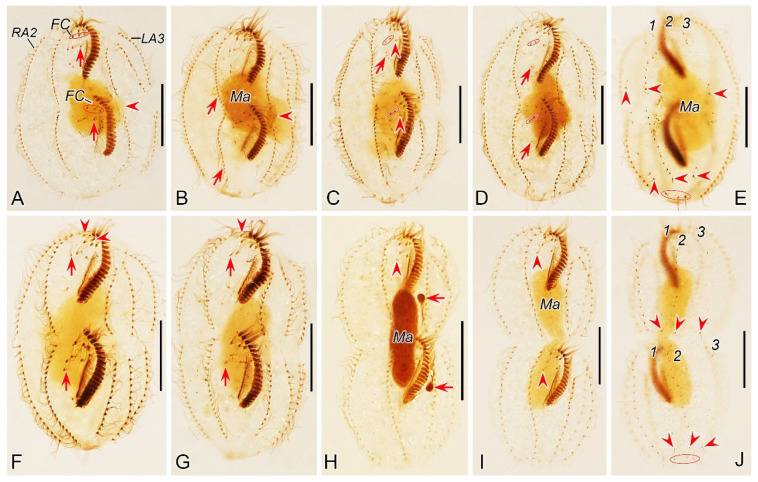
Morphogenetic photomicrographs of *Perisincirra paucicirrata* after protargol staining. (**A**) Ventral view of an early-middle divider, arrows mark buccal cirrus of the proter and opisthe, arrowhead indicates the dividing micronucleus, in this stage, the macronuclear nodules fuse into a single mass. (**B**) Ventral view of a middle divider, showing anlagen I–IV differentiating into cirri, arrows point to the completion of development of marginal rows anlagen and arrowhead indicates the dividing micronucleus. (**C**,**D**) Ventral views of middle dividers, arrowheads mark buccal cirrus, arrows indicate anlage IV and the cirrus from it, which will be resorbed later, ellipses encircle parabuccal cirri formed from anlagen III and IV. (**E**,**F**) Dorsal (**E**) and ventral (**F**) views of the same late-middle divider, arrowheads in (**E**) showing the new caudal cirri formed at the rear end of dorsal kineties anlagen, ellipse encircles the parental caudal cirri, arrows in (**F**) indicate the parabuccal cirri formed from the anlage IV, arrowheads in (**F**) mark the parental frontal cirri, in this stage, the left parental frontal cirrus resorbed. (**G**) Ventral view of a late-middle divider, arrows indicate the parabuccal cirri formed from the anlage IV, arrowhead demonstrates the right parental frontal cirrus, up to this stage, the left and middle parental frontal cirri resorbed. (**H**) Ventral view of a late divider, showing the posterior parabuccal cirrus of the proter and opisthe migrate leftward and appear below the anterior parabuccal cirrus (arrowhead), arrows denote micronuclei, in this stage, the micronucleus divides into two mitotically and distributes to the proter and the opisthe; the macronuclear mass is ready for the first round of division. (**I**,**J**) Ventral (**I**) and dorsal (**J**) views of the same late divider, arrowheads in (**I**) mark the posterior parabuccal cirrus of the proter and opisthe; arrowheads in (**J**) indicate the caudal cirri for the proter and opisthe, ellipse encircles the parental caudal cirri, in this stage, the macronuclear nodule completes its first division. 1–3, dorsal kineties anlagen 1–3; FC, frontal cirri; LA3, outer left marginal row anlage; Ma, macronuclear nodules; RA2, outer right marginal row anlage. Scale bars = 30 μm.

**Figure 10 microorganisms-12-02013-f010:**
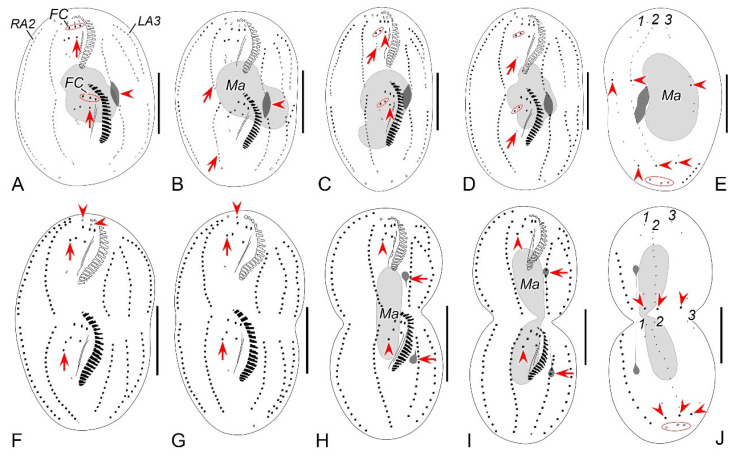
Middle to late morphogenetic stages of *Perisincirra paucicirrata* after protargol staining. (**A**) Ventral view of an early-middle divider, arrows mark buccal cirrus of the proter and opisthe, arrowhead indicates the dividing micronucleus, in this stage, the macronuclear nodules fuse into a single mass. (**B**) Ventral view of a middle divider, showing anlagen I–IV differentiating into cirri, arrows point to completion of development of marginal rows anlagen and arrowhead indicates the dividing micronucleus. (**C**,**D**) Ventral views of middle dividers, arrowheads mark buccal cirrus, arrows indicate anlage IV and the cirrus from it, which will be resorbed later, ellipses encircle parabuccal cirri formed from anlagen III and IV. (**E**,**F**) Dorsal (**E**) and ventral (**F**) views of the same late-middle divider, arrowheads in (**E**) showing the new caudal cirri formed at the rear end of dorsal kineties anlagen, ellipse encircles the parental caudal cirri, arrows in (**F**) indicate the parabuccal cirri formed from the anlage IV, arrowheads in (**F**) mark the parental frontal cirri, in this stage, the left parental frontal cirrus resorbed. (**G**) Ventral view of a late-middle divider, arrows indicate the parabuccal cirri formed from the anlage IV, arrowhead demonstrates the right parental frontal cirrus, up to this stage, the left and middle parental frontal cirri resorbed. (**H**) Ventral view of a late divider, showing the posterior parabuccal cirrus of the proter and opisthe migrate leftward and appear below the anterior parabuccal cirrus (arrowhead), arrows denote micronuclei, in this stage, the micronucleus divides into two mitotically and distributes to the proter and the opisthe; the macronuclear mass is ready for the first round of division. (**I**,**J**) Ventral (**I**) and dorsal (**J**) views of the same late divider, arrowheads in (**I**) mark the posterior parabuccal cirrus of the proter and opisthe, arrows point to micronuclei; arrowheads in (**J**) indicate the caudal cirri for the proter and opisthe, ellipse encircles the parental caudal cirri, in this stage, the macronuclear nodule completes its first division. 1–3, dorsal kineties anlagen 1–3; FC, frontal cirri; LA3, outer left marginal row anlage; Ma, macronuclear nodules; RA2, outer right marginal row anlage. Scale bars = 30 μm.

**Figure 11 microorganisms-12-02013-f011:**
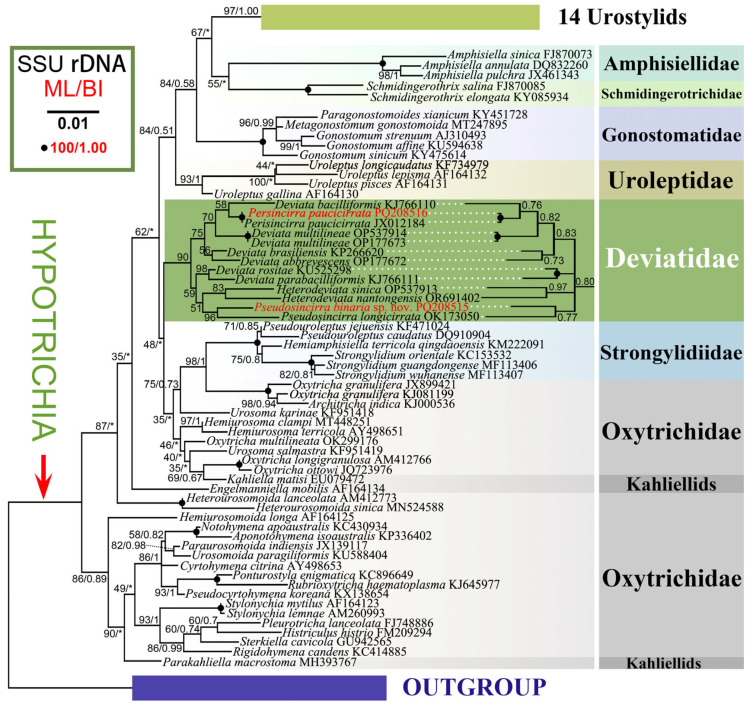
Maximum likelihood (ML) tree based on the SSU rRNA gene sequences showing the positions of the two newly sequenced species (in red). Numbers at the nodes represent the ML bootstrap support and BI posterior probability values. Fully supported (100/1.00) branches are marked with filled circles. Symbol “Asterisks” indicate disagreement between the BI tree and the reference ML tree. All branches are drawn to scale. The scale bar corresponds to one substitution per 100 nucleotide positions.

**Table 1 microorganisms-12-02013-t001:** Morphometric characterization of *Pseudosincirra binaria* sp. nov.

Character	Min	Max	Mean	M	SD	CV	n
Body length	145	211	174	170	19.7	11.3	25
Body width	42	131	78	75	20.6	26.3	25
Body length:body width, ratio	1.5	3.8	2.3	2.3	0.5	21.8	25
Adoral zone length	35	48	41.4	41.3	3.2	7.7	25
Adoral zone length: body length, ratio (%)	19	32	24	24.3	3.0	12.4	25
Adoral membranelles, number	20	25	23	23	1.2	5.1	25
Macronuclear nodules, number	4	4	4.0	4	0	0	25
Macronuclear nodules, length	12	18	14.1	14.0	1.6	11.7	25
Macronuclear nodules, width	5	9	6.8	6.4	1.1	16.3	25
Micronuclei, number	1	2	2.0	2	0.2	10.2	25
Micronuclei, diameter	4	6	4.7	4.7	0.5	9.7	25
Frontal cirri, number	3	3	3	3	0	0	25
Buccal cirrus, number	1	1	1	1	0	0	25
Parabuccal cirri, number	1	1	1	1	0	0	25
Frontoventral row, number	1	1	1	1	0	0	25
Frontoventral cirri, number	15	21	18	18	1.6	8.6	25
Left marginal rows, number	3	3	3	3	0	0	25
Cirri in L1, number	14	19	16.7	17	1.5	8.8	25
Cirri in L2, number	14	18	16.2	16	1.1	7.0	25
Cirri in L3, number	14	19	16.5	17	1.3	8.0	25
Right marginal rows, number	2	2	2	2	0	0	25
Cirri in R1, number	20	26	23.2	23	1.7	7.4	25
Cirri in R2, number	21	26	23	23	1.5	6.6	25
Dorsal kineties, number	3	3	3	3	0	0	25
Dikinetids in anterior DK1, number	4	6	5.2	5	0.7	13.3	25
Dikinetids in posterior DK1, number	3	5	3.8	4	0.5	13.9	25
Dikinetids in DK1, number	7	10	8.9	9	1.0	10.7	25
Dikinetids in DK2, number	16	20	17.7	18	1.4	7.8	25
Dikinetids in DK3, number	12	16	13.6	13	1.2	8.8	25
Caudal cirri, number	3	3	3	3	0	0	25

Measurements in µm. Data based on protargol-stained specimens. AZM, adoral zone of membranelles; CV, coefficient of variation in %; DK1−3, dorsal kineties 1−3; L1, inner left marginal row; L2, middle left marginal row; L3, outer left marginal row; M, median; Max, maximum; Mean, arithmetic mean; Min, minimum; n, number of specimens measured; R1, inner right marginal row; R2, outer right marginal row; SD, standard deviation.

**Table 2 microorganisms-12-02013-t002:** Morphometric characterization of *Perisincirra paucicirrata*.

Character	Min	Max	Mean	M	SD	CV	n
Body length	75	135	105	108	16.2	15.4	21
Body width	32	58	42	42	7.6	18.3	21
Body length:both width, ratio	2.2	3	2.5	2.5	0.2	8	21
Length of adoral zone	22	35	27.7	28.7	3.3	11.8	21
AZM: body length, ratio	21	33	26.6	25.7	3.1	11.7	21
Adoral membranelles, number	18	20	19	19	0.7	3.5	21
Macronuclear nodules, number	2	2	2	2	0	0	21
Macronuclear nodules, length	19	36	27.5	28.1	5.8	21.2	21
Macronuclear nodules, width	8	18	12.4	11.9	2.9	23.3	21
Micronuclei, number	1	1	1	1	0	0	21
Micronuclei, diameter	4	7	5	5.2	1.0	20.2	21
Frontal cirri, number	3	3	3	3	0	0	21
Buccal cirrus, number	1	1	1	1	0	0	21
Parabuccal cirri, number	2	2	2	2	0	0	21
Left marginal rows, number	3	3	3	3	0	0	21
Cirri in L1, number	12	16	13.5	13	1.0	7.6	21
Cirri in L2, number	11	16	12.9	13	1.2	9.3	21
Cirri in L3, number	11	18	14	14	1.7	12.0	21
Right marginal rows, number	2	2	2	2	0	0	21
Cirri in R1, number	18	23	20.3	20	1.4	7.0	21
Cirri in R2, number	17	22	19.2	19	1.5	8	21
Dorsal kineties, number	3	3	3	3	0	0	21
Dikinetids in DK 1, number	2	4	3.3	3	0.6	17.1	21
Dikinetids in DK 2, number	8	11	9.8	10	0.9	9.1	21
Dikinetids in DK 3, number	3	4	3.3	3	0.5	14.1	21
Caudal cirri, number	3	3	3	3	0	0	21

Measurements in µm. All data based on protargol-stained specimens. AZM, adoral zone of membranelles; CV, coefficient of variation in %; DK, dorsal kineties; L1, inner left marginal row; L2, middle left marginal row; L3, outer left marginal row; M, median; Max, maximum; Mean, arithmetic mean; Min, minimum; n, number of specimens measured; R1, inner right marginal row; R2, outer right marginal row; SD, standard deviation.

**Table 3 microorganisms-12-02013-t003:** Comparison of 18S rRNA gene sequences similarity between deviatids.

Scheme	Sequence Similarity												
	1	2	3	4	5	6	7	8	9	10	11	12	13
1 *Pseudosincirra binaria*	-	25	25	30	25	25	24	23	26	33	34	34	26
2 *Perisincirra paucicirrata*	0.988	-	0	18	19	19	12	18	22	27	32	22	33
3 *Perisincirra paucicirrata*	0.985	0.987	-	18	19	19	12	18	22	27	32	22	33
4 *Deviata bacilliformis*	0.985	0.987	1.000	-	23	23	26	28	24	31	37	32	37
5 *Deviata multilineae*	0.983	0.992	0.989	0.989	-	0	16	19	21	28	36	34	31
6 *Deviata multilineae*	0.981	0.988	0.987	0.987	0.994	-	16	19	21	28	36	34	31
7 *Deviata brasiliensis*	0.984	0.985	0.986	0.986	0.986	0.985	-	9	21	28	30	30	31
8 *Deviata abbrevescens*	0.979	0.982	0.981	0.981	0.981	0.981	0.990	-	22	29	31	33	32
9 *Deviata rositae*	0.976	0.979	0.976	0.976	0.980	0.979	0.979	0.977	-	15	31	27	30
10 *Deviata parabacilliformis*	0.979	0.985	0.977	0.977	0.980	0.978	0.982	0.979	0.980	-	34	32	37
11 *Heterodeviata sinica*	0.976	0.978	0.979	0.979	0.979	0.979	0.980	0.975	0.977	0.976	-	30	35
12 *Heterodeviata nantongensis*	0.980	0.983	0.983	0.983	0.984	0.985	0.983	0.978	0.977	0.977	0.983	-	36
13 *Pseudosincirra longicirrata*	0.988	1.000	0.987	0.987	0.992	0.988	0.985	0.982	0.979	0.985	0.978	0.983	-

Each matrix shows the percentage of sequence identity (below the diagonal) and the number of unmatched nucleotides (above the diagonal). The GenBank accession number from 1 to 13 are PQ208515, PQ208516, JX012184, KJ766110, OP537914, OP177673, KP266620, OP177672, KU525298, KJ766111, OP537913, OR69140 and OK173050, respectively.

**Table 4 microorganisms-12-02013-t004:** Comparison of Qingdao population of *Perisincirra paucicirrata* with other populations and congener.

Characters	*P. paucicirrata*	*P. paucicirrata*	*P. paucicirrata*	*P. kahli*
Body length, in vivo	75–125	60–110	70–130	85–160
Body length:width, ratio	4.7:1	4:1	5:1	10–15:1
Frontal cirri, number	3	3	3	3
Ma, number	2	2	2	2
Mi, number	1	1	1	2
AM, number	18–20	16–19	13–17	18–20 *
Parabuccal cirri, number	2	1–3	1–3	–
Right marginal rows, number	2	2	2	2
Cirri in RMR 1, number	18–23	15–24	5–12	–
Cirri in RMR 2, number	17–22	14–23	4–8	18–20
Left marginal rows, number	3	3–4	2	2
Cirri in LMR 1, number	12–16	11–15	3–9	
Cirri in LMR 2, number	11–16	10–16	5–8	16–20
Cirri in LMR 3, number	11–18	11–17	–	–
Dorsal kineties, number	3	3	3	3
Caudal cirri, number	3	3	3	–
Data source	present study	[36]	[9,20]	[20]

All measurements are in micrometres. AM, adoral membranelles; LMR, left marginal rows; Ma, macronuclear nodules; Mi, micronuclei; RMR, right marginal rows; –, data not available. * Data from drawing in [56].

## Data Availability

The data presented in the study are deposited in the GenBank database (https://www.ncbi.nlm.nih.gov/genbank, accessed on 28 June 2024), accession number: PQ208515 (*Pseudosincirra binaria* sp. nov.) and PQ208516 (*Perisincirra paucicirrata*).

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
