# Peer review of "Morphology, Morphogenesis and Molecular Phylogeny of Two Freshwater Ciliates (Alveolata, Ciliophora), with Description of Pseudosincirra binaria sp. nov. and Redefinition of Pseudosincirra and Perisincirra"

_microorganisms, 2024, doi:10.3390/microorganisms12102013_

Round 1

Reviewer 1 Report

Comments and Suggestions for Authors

This ms presents a discussion of the new species Pseudosincirra binaria and redefinition of Pseudosincirra and Perisincirra using a polyphasic approach based on morphology, morphogenesis and molecular phylogenetic analysis. The descriptions are clear and the images of good quality, and, after comparing their descriptions with previously published descriptions of the genera and species, I agree with their conclusions. I recommend publication after minor revision.

Particular suggestion:

1.     The Materials and Method  “Measurements and counts on stained specimens were carried out at a magnification of 1,000X.” Ciliates are relatively large and I don't think they need such a large magnification.

2.     “Based on morphological study mentioned above, single cells of each species were isolated from a raw culture (without other deviated species recognized) and underwent five washes with sterilized habitat water to eliminate potential contaminations.” Can sterilized habitat water be replaced with sterilized bi-distilled water?

3.     The type locality  Do you have more information about the physical and chemical parameters?

4.     “3.3.18. S rRNA Gene Sequences and Phylogenetic Analyses” change to“3.3.1.1”?

5.     When you sequenced the 18S rRNA gene, did you repeat it?

6.     ML tree  Why do you choose four outgroup species in a phylogenetic trees?

        In phylogenetic trees, the closer the species are, the more similar they are. However, this is not the case in the ms. Please confirm the relationship and similarity of the species in the phylogenetic tree again.

        Please add a table comparing the 18S rRNA sequence similarity between the described species and its closed genera.

7.     The discussion  Genus and species names should be italicized. Please double-check the writing of the terms in the full ms.

Comments on the Quality of English Language

Some minor revision are needed

Reviewer 2 Report

Comments and Suggestions for Authors

Overall, this is a very nicely organized manuscript and contains substantial light microscopic and molecular genetic analyses to support the taxonomic revisions and new species presented. This particular taxonomic group of ciliates is not my area of expertise, but I can comment on the quality of the scientific approach and interpretations. The combination of morphological, cytological and molecular genetic evidence is particularly strong, and the authors have taken care to strengthen their taxonomic interpretations by including developmental stages of the ciliate taxa in addition to documenting the ciliature and micromorphological features of mature individuals. 

I have mainly minor suggestions regarding presentation of the text.

Page    Line     Comment

7          8          Is it possible for the authors to be somewhat more exact about the rate of locomotion?   “---- moderately rapid gliding,-----” is not very informative unless the reader has had a substantial amount of experience in observing different ciliate locomotion.  If the authors can make some estimates of rate of locomotion more quantitatively it would be helpful.

21        Fig. 11             The use of red font on the dark green background in the molecular genetic tree to identify the species names of the two newly sequenced ciliates is impossible to read against the dark green background, at least in my version of the manuscript.  It may be more legible if yellow or some color with less density is used.

22-23   Section 4.2.  There are several species names in this section that are printed in normal font and need to presented in italics. I am confident the authors will see them if they read through this section.
